# Development of a Sham Protocol to Investigate Transcutaneous Tibial Nerve Stimulation in Randomised, Sham-Controlled, Double-Blind Clinical Trials

**DOI:** 10.3390/biomedicines11071931

**Published:** 2023-07-07

**Authors:** Stephanie A. Stalder, Stéphanie van der Lely, Collene E. Anderson, Veronika Birkhäuser, Armin Curt, Oliver Gross, Lorenz Leitner, Ulrich Mehnert, Martin Schubert, Jure Tornic, Thomas M. Kessler, Martina D. Liechti

**Affiliations:** 1Department of Neuro-Urology, Balgrist University Hospital, University of Zürich, 8008 Zürich, Switzerland; 2Department of Health Sciences and Technology, ETH Zürich, 8092 Zürich, Switzerland; 3Swiss Paraplegic Research, 6207 Nottwil, Switzerland; 4Department of Health Sciences and Medicine, University of Lucerne, 6002 Lucerne, Switzerland; 5Spinal Cord Injury Centre, Balgrist University Hospital, University of Zürich, 8008 Zürich, Switzerland

**Keywords:** tibial nerve stimulation, transcutaneous tibial nerve stimulation, randomised controlled trials, double-blind, sham stimulation, sham development, sham protocol, electrical stimulation

## Abstract

Transcutaneous tibial nerve stimulation (TTNS) is a promising treatment for neurogenic lower urinary tract symptoms. However, the evidence is limited due to a general lack of randomised controlled trials (RCTs) and, also, inconsistency in the sham and blinding conditions. In the context of much-needed RCTs, we aimed to develop a suitable sham-control protocol for a clinical setting to maintain blinding but avoid meaningful stimulation of the tibial nerve. Three potential electrode positions (lateral malleolus/5th metatarsal/plantar calcaneus) and two electrode sizes (diameter: 2.5 cm/3.2 cm) were tested to determine which combination provided the optimal sham configuration for a TTNS approach, based on a visible motor response. Sixteen healthy volunteers underwent sensory and motor assessments for each sham configuration. Eight out of them came back for an extra TTNS visit. Sensory thresholds were present for all sham configurations, with linear regression models revealing a significant effect regarding electrode position (highest at plantar calcaneus) but not size. In addition, motor thresholds varied with the position—lowest for the 5th metatarsal. Only using this position and 3.2 cm electrodes attained a 100% response rate. Compared to TTNS, sensory and motor thresholds were generally higher for the sham configurations; meanwhile, perceived pain was only higher at the lateral malleolus. In conclusion, using the 5th metatarsal position and 3.2 cm electrodes proved to be the most suitable sham configuration. Implemented as a four-electrode setup with standardized procedures, this appears to be a suitable RCT protocol for maintaining blinding and controlling for nonspecific TTNS effects in a clinical setting.

## 1. Introduction

Tibial nerve stimulation (TNS) is an effective and safe option for the treatment of lower urinary tract symptoms (LUTS) [1,2,3]. This form of neuromodulation involves the use of electrical impulses to target the innervation system of the lower urinary tract via the tibial nerve [4]. Stimuli can be applied either percutaneously (PTNS), using needle electrodes [3,5], or transcutaneously (TTNS), using surface electrodes [6].

Patients with neurological diseases were previously discouraged from TNS, using the argument of impaired stimulation pathways. Nevertheless, a few studies [2] indicate that TNS may also work in patient groups with neurogenic lower urinary tract dysfunction (NLUTD), such as, multiple sclerosis [7,8], Parkinson’s disease [9,10,11], and spinal cord injury [12,13]. This makes sense under the assumption that neuromodulation is actually addressing the impaired nervous system by influencing neuronal circuit patterns relevant to the function of the lower urinary tract. Recent evidence from a randomised controlled trial (RCT) showed that patients with NLUTD improved after sacral neuromodulation [14]. For TNS applications, however, we need more robust evidence from high-quality RCTs before we can support their implementation into routine clinical practice for patients with neurological diseases.

As far as previous RCTs are concerned, different results and control conditions have been reported, which could, in part, be attributable to the use of heterogenous TNS administration routes and control protocols. The effectiveness of PTNS was shown in several RCTs via using a sham treatment [15,16] or being analysed in comparison to medical treatment [17,18]. The transcutaneous application (TTNS) is still less common; although, it is more attractive, considering its non-invasiveness, possibility for home application, and lower costs [1,2,19,20]. There are only a few RCTs and they provide limited evidence for TTNS’s effectiveness in patients with lower urinary tract dysfunction (LUTD) [12,21,22]. There is, however, inconsistency in the control configuration (sham) regarding electrode placement, electrode size, procedures, and blinding setup, which hampers the interpretability and only allows partial control for nonspecific TTNS effects [1,2,12,16,22]. In these previous studies, sham configurations were chosen to avoid or minimize the current applied to the tibial nerve. While one study placed the electrodes at the little toe of the contralateral foot [16], a study by Booth et al. stimulated at the lateral malleolus [6]. Another attempt using a subsensory stimulation (for TTNS and sham) was explored in our recent pilot study of nine patients with refractory NLUTD [23]. While this approach ensured blinding maintenance, the reduced stimulation intensity raised the question of whether patients received sufficient stimulation for clinical improvements; hence, the subsensory approach was not further pursued. Similarly, the attempt by Perissinotto et al. of using a sham configuration without applying any electrical current to the patients and conducting no initial assessment of the correct electrode placement seems less promising as this procedure deviated from the TTNS approach and could lead to reduced sensory input compared to standard TTNS [22]. This has been solved by applying partial stimulation at the standard TTNS location, for example, by providing a comparable sensation during the initial determination of the thresholds [12]. However, with such an approach, the sham group also receives a TTNS treatment dose, which, when accumulated over many sessions, may induce partial TTNS effects.

Theoretically, an active sham stimulation, spatially distant from the target (i.e., tibial) nerve, e.g., at the upper extremity, would be tempting. However, patients could easily figure out whether or not they received stimulation at the tibial nerve. It is another matter of whether such a sham stimulation might even have a positive impact on LUT function. Considering sensory perception, the stimulation should be administered (at least briefly) at or near the TTNS location (medial malleolus), meaning somewhere on the foot. In terms of aiming for a most distant location, the plantar calcaneus may be suitable. Another factor relevant to the effectiveness and comfort of the TNS is the electrode type and size. The choice of electrode size depends on several factors, such as the desired treatment area, precision of stimulation, patient comfort, and individual anatomical variation. Proper electrode placement and the consideration of these factors can help to optimize the effectiveness and comfort of TNS therapy. For PTNS, typically, a needle is used as the stimulation electrode and a round-surface electrode is placed as a second electrode. The needle approach is minimally invasive; however, it allows for the application of current close to the nerve, requiring minimal current to elicit the desired motor response. In contrast, TTNS utilizes larger surface electrodes (areas of several square centimetres) to enhance the recruitment of the tibial nerve; however, higher currents are required to invoke the respective motor response. Various sizes and shapes of surface electrodes have been utilized in different TTNS studies; however, specific information regarding the exact dimensions of these electrodes is often lacking. For sham conditions, mostly surface electrodes have been used. To provide a similar level of stimulation, the sham electrodes should have approximately the same size as those used for TTNS. Another factor to be taken into account is the stimulation frequency for TNS, which, based on the literature, ranges from 10 to 20 Hz [6,21,24,25,26]. A recent review by Li et al. [27] identified 20 Hz as the most commonly used frequency for TNS. In any case, it is important to match the stimulation settings between sham and TTNS for well-controlled studies. Taking all of the above factors into account, the optimal TTNS sham-control configuration is yet to be determined.

The overall aim of our research efforts was to develop an optimal sham configuration for upcoming RCTs investigating the effectiveness of TTNS in a clinical setting. In the envisaged RCTs, a 20 Hz stimulation approach with two surface electrodes (3.2 cm) over the tibial nerve [23] shall be used for TTNS—henceforth, referred to as “verum”, a common Latin label for “which is true” [28]. For verum TTNS, a preparation period is necessary in which the correct placement of electrodes is indicated through the visibility of a motor response, i.e., big toe flexion [12]. For the intervention period, the stimulation intensity will be set relative to motor (instead of sensory) thresholds, which is the clinical standard for TNS. This ensures patients will receive a sufficient dose for clinical improvement. The desired sham setup and procedures should match those of the utilised verum TTNS to control for nonspecific treatment effects and to keep the subjects blind to the treatment allocation (verum or sham). Considering the literature and our considerations, we tested three potential electrode positions and two electrode sizes (with different diameters/areas through which the current is transmitted to the skin) to determine which sham configuration triggers optimal motor responses and perceived sensations.

## 2. Materials and Methods

Advancing our previous study [23], this methodological development was conducted as part of the preparation period of two randomised, sham-controlled, double-blind clinical trials: Transcutaneous tibial nerve stimulation in patients with Acute Spinal Cord Injury to prevent neurogenic detrusor overactivity: a nationwide randomized, sham-controlled, double-blind clinical trial (TASCI, ClinicalTrials.gov: NCT03965299, [29]) and bladder and TranscUtaneous tibial Nerve stimulation for nEurogenic lower urinary tract Dysfunction (bTUNED, Clinical Trials.gov: NCT04315142, [30]). The desired sham protocol should be designed to match the verum TTNS. Additionally, the control configuration should be optimised regarding blinding, experimental setup, and procedures to avoid meaningful stimulation of the tibial nerve.

Based on the literature, three different potential sham positions and two electrode sizes (included in our stimulator kit) were evaluated regarding the presence of motor responses (response rate—RR) and perceived sensations in healthy subjects. In order to provide reference values as comparators for the sham assessments, additional verum TTNS assessments were performed on half of the participants during a separate visit. To ensure a balanced distribution of testing sequences across the population in terms of position and electrode size, the subjects were assigned using a pseudo-randomised list.

In view of the upcoming RCTs with repeated intervention sessions, a first pilot was conducted in a hospital setting with a patient who was representative of the TTNS target population. For this, the most suitable among the three investigated sham positions was incorporated into a four-electrode setup. Over several weeks the patient had stimulation sessions; each session was randomly assigned to either verum or sham TTNS and sensory and motor thresholds were assessed.

This study was approved by the local ethics committee (#2019-00074). Study data were collected and managed using REDCap (Research Electronic Data Capture) electronic data capture tools hosted at the Balgrist University Hospital [31,32].

### 2.1. Study Population

Healthy subjects were recruited and had to fulfil predefined criteria for study inclusion: no neurological disease, no regular intake of medication (except contraceptives), a Montreal Cognitive Assessment [33] (MoCA) score ≥ 26, and a Hospital Anxiety and Depression Scale [34] (HADS) value of ≤7 for anxiety and depression subscores. The required health status was defined as the absence of any health troubles or pain, as assessed by sensory testing and standardized medical history taking, including a medication list. For the first application of the procedure a patient representative of the target population of the TASCI trial was recruited. This patient had to fulfil the following criteria: spinal cord injury (SCI) with some residual sensory function and undergoing primary rehabilitation in a hospital setting.

### 2.2. Stimulation/Intervention

The stimulation was performed with a commercially available electrical nerve stimulation device (ELPHA II 3000, CE 0543 Certification, FH Service, Odense, Denmark) using round adhesive gel electrodes (PALS Neurostimulation Electrodes, provided in the ELPHA stimulator kit, Axelgaard Mfg. Co., Lystrup, Denmark) with a 2.5 cm or 3.2 cm diameter, respectively. The stimulation parameters were set to a frequency of 20 Hz and a pulse width of 0.2 ms, in accordance with the chosen TTNS parameters for the verum configuration [23]. The current amplitude could be manually adjusted between 0 mA and 100 mA in steps of 0.5 mA.

For the three sham positions under investigation, the electrodes were placed at:(a)The lateral malleolus [6], with one electrode placed 4–5 cm proximal and 4–5 cm posterior to the lateral malleolus and the second electrode placed 3–4 cm proximal to the first one (Figure 1a);(b)The 5th metatarsal [16], with one electrode placed on the dorsal forefoot, over the 5th metatarsal, and the second electrode placed on the plantar forefoot, under the 5th metatarsal on the fat pad (Figure 1b);(c)The plantar calcaneus, with two electrodes placed centrally 1–2 cm apart on the plantar calcaneus (Figure 1c).

For verum TTNS, two electrodes were utilized. The first electrode was placed approximately 4–5 cm proximal and 4–5 cm posterior to the medial malleolus while the second electrode was placed in the middle of the medial longitudinal arch of the foot [23] (Figure 2).

### 2.3. Procedures

During the first session, the subjects’ health statuses was determined using standardized questionnaires (HADS, MoCA) and their medical histories being taken by qualified study personnel. To confirm intact sensory function, pinprick, light touch, vibration, and thermal testing (cold and warm sensation) were assessed at the distal limbs (sensory testing). The subjects were asked to lie in a comfortable and supine position on an examination table with extended legs in order to be free from any nerve compression at the knee joint [20]. A pillow was placed on the lower part of each subjects’s leg to restrict the subjects’ view of their foot. The leg side was randomly assigned—the right leg was used in eight subjects. Each stimulation session started with a preparatory period, during which the skin was first degreased using an alcoholic solution and was abraded additionally by using an abrasive gel to improve the conductivity between the electrodes and the skin at all positions under investigation. Three sham positions (Figure 1) and two different electrode diameters were compared.

For the first round of testing, a pair of equally sized electrodes were placed at every sham position (a total of six electrodes with a diameter of 2.5 cm, or 3.2 cm, respectively). The three different electrode positions were investigated in sequence, according to the assigned order of stimulation (randomised among the three positions and two electrode sizes). The electrodes for the first position were connected to the ELPHA II 3000 device and the current amplitude was slowly increased (approximately 1 mA/s) (Figure 3). The subjects were asked to report the first sensation, which was defined as the sensory threshold (current perception threshold—CPT; throughout the manuscript coloured blue) using the methods of limits [35]. In addition, the subjects were asked to describe their sensation (localisation of the stimulation and what they felt, e.g., tingling, prickling). The stimulus intensity was further increased in order to see if a motoric response with contraction/movement of the foot muscles could be provoked, i.e., the motor threshold (MT; throughout the manuscript coloured red) for this specific electrode placement. The MT was further characterized regarding the triggered movement (toe fanning, muscle twitch, pulsation) and corresponding localization (e.g., first, second toe). In addition, the subjects indicated whether the stimulation was painful at the time of the motor response using a Numerical Rating Scale (NRS) ranging from 0 to 10 in which 0 = “no pain” and 10 = “worst pain imaginable” [36]. As a next step, the stimulus intensity was gradually decreased until the subjects reported the cessation of the sensation (current cessation threshold—CCT; throughout the manuscript coloured green). To ensure the reliability of the responses, the sensory and motor assessments were replicated (a minimum of two times, possibly more, until consistent and reliable values were obtained). Following this, the electrodes from the second sham position (according to the randomised order) were connected to the stimulator and the same standardised procedure as described above was performed for the second and third sham positions (Figure 1).

In a second stimulation round, the electrodes of the other sizes—i.e., different diameters/areas through which the current is transmitted to the skin—were tested. For this, another three pairs of electrodes with matching diameters (2.5 cm or 3.2 cm, respectively) were placed on the same three sham positions at each subject’s foot. The procedure was repeated in the same manner as for the first round. For verum TTNS assessments (extra visit; for setup, see Figure 2), the sensory and motor thresholds were evaluated by using the same leg as that used for the sham stimulations using 2.5 cm and 3.2 cm electrodes, as described above.

### 2.4. Data Analysis and Statistics

For repeated measurements, such as sensory thresholds, the medians per subject were determined and used for group statistics, which were presented as boxplots. Sensory threshold and motor response characteristics (i.e., threshold stimulation intensities, NRS) were summarized using descriptive statistics (e.g., group median, lower whisker/upper whisker (maximum 1.5 × interquartile range), 25th percentile, 75th percentile). The response rate (RR) for sensory thresholds depicts the percentage of perceived stimulations among all healthy subjects. Regarding motor response, PR reflect the percentage of assessments showing a motor response.

For each generalized linear models (GLMs), we defined the CPT, CCT, MT, or NRS as the dependent variable and used electrode size, as well as electrode position (for sham only), as the independent variables. Post hoc pairwise comparisons of categorical variable levels, corrected for multiple testing, were performed, if necessary, using the “emmeans” package in RStudio (R Statistical Software v4.2.2 (Boston, MA, USA)). Regarding the differences in the thresholds and NRS scores between verum TTNS and the different sham positions, Wilcoxon signed-rank tests were used to compare each sham position to verum TTNS. All data were tested for normal distributions using Shapiro–Wilk tests and visual inspections of histograms and quantile–quantile (Q–Q) plots. Statistical significance was set at α < 0.05.

### 2.5. First Patient Application

For the first application with a patient, a four-electrode setup was used, allowing for the application of either verum or sham TTNS without changing the outer stimulation setup. For this, two electrodes were placed at the sham position, which was considered the most suitable based on our results from the sham assessments in healthy subjects (see Section 3.1). Two additional electrodes were placed at the medial malleolus (Figure 2). Over several weeks of randomly assigned verum or sham TTNS sessions, sensory and motor assessments were performed following the same procedure as outlined above for the healthy subjects. For each 30-minute intervention, the stimulation amplitude was reduced to a submotor level, defined as 0.5–1.0 mA below the motor threshold.

For data analysis, the median values of the sensory and motor thresholds were computed per session and for the repeated measurements collected over several weeks. Descriptive statistics, such as median, 25th percentile, and 75th percentile, were employed to summarize sensory threshold and motor response characteristics. The RRs for both sensory and motor thresholds were calculated separately for the verum and sham TTNS conditions.

## 3. Results

Sixteen healthy subjects (eight females) with an average age of 31 years (range of 25–43 years) were included. Eight subjects (average age of 31 years, range of 25–43 years) came back for a second visit for verum TTNS assessments. The pilot patient was a 59-year-old female with a traumatic incomplete SCI (lesion level T9, American Spinal Injury Association Impairment Scale (AIS) C [37]), 50 days after SCI at study start, lower extremity motor score: 22/50, light touch: 98/112, pinprick: 98/112).

### 3.1. Comparison of Different Sham Positions and Electrode Sizes

Analyses of sensory thresholds revealed 100% RRs for CPT and CCT, irrespective of position and electrode size (Table 1). For motor responses, the RRs differed between positions and electrode sizes (Table 1). The MT response rate was the lowest with the stimulation at the plantar calcaneus (44% (7/16 subjects) for the 2.5 cm electrodes; 75% (12/16 subjects) for the 3.2 cm electrodes). Higher RRs were observed at the lateral malleolus (75% (12/16 subjects) for the 2.5 cm electrodes; 88% (14/16 subjects) for the 3.2 cm electrodes). A 100% RR was only found at the 5th metatarsal position using 3.2 cm electrodes (Table 1). Stimulation at this position induced motor responses from the 3rd, 4th, and 5th toes.

Well-tolerable sensory responses, as well as non-painful motor responses (0 on the NRS scale), were found when using the 5th metatarsal sham position in all subjects (median NRS = 0; 25th percentile = 0; 75th percentile = 0.63) and in 86% of the subjects for the lateral malleolus position (median NRS = 1; 25th percentile = 0.25; 75th percentile = 2.25).

**Figure 4 biomedicines-11-01931-f004:**
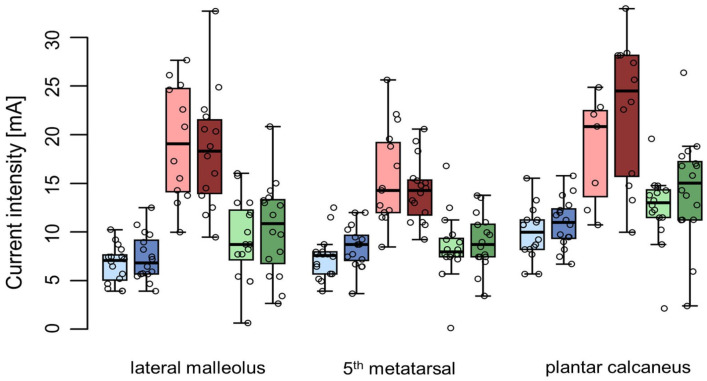
Boxplots for current perception threshold (CPTs—blue), motor threshold (MTs—red), and current cessation threshold (CCTs—green) values (n = 16) for 2.5 cm electrodes (light colours, left bars) and 3.2 cm electrodes (dark colours, right bars) stratified in the sham position. Box–Whisker plot indications: lower whisker/upper whisker (maximum 1.5 × interquartile range), 25th percentile, 75th percentile, and median current intensities [mA].

GLM regression across all subjects revealed no significant effects of electrode size on CPT, CCT, MT (Figure 4), and NRS score. However, there was an effect due to electrode position: significantly higher sensory thresholds were noted for the plantar calcaneus sham position compared to the lateral malleolus and to the 5th metatarsal sham position. Regarding the MT, the 5th metatarsal sham position showed significantly lower thresholds compared to the lateral malleolus (Table 2). Regarding the NRS, the lateral malleolus position showed significantly higher values compared to the 5th metatarsal position.

### 3.2. Sham Measurements in Comparison to Verum TTNS Assessments

For the comparison with verum TTNS, only the 3.2 cm sham electrodes were considered due to the higher motor RRs compared to those associated with the smaller electrodes. The 5th metatarsal and the plantar calcaneus sham positions had higher CPTs compared to that of verum TTNS. Regarding MT and CCT, all sham positions demonstrated higher values compared to verum TTNS, whereby the 5th metatarsal tended to be the closest to verum TTNS (Figure 5).

Regarding perceived pain at the intensities required for a motor response, stimulation at the lateral malleolus was perceived to be slightly more painful compared to the verum TTNS at the medial malleolus. The perceived pain was the lowest at the 5th metatarsal sham position when using the 3.2 cm electrodes (NRS median 0; range 0–1.5) (Figure 5). For verum TTNS, the size (3.2 cm vs. 2.5 cm) of the electrodes did not have any significant effect on CPT, CCT, MT, and NRS score in the GLM and there was a 100% RR for all outcomes among both electrode sizes.

Overall, our research showed the best results for the 5th metatarsal position; it is the most suitable sham-control configuration for TTNS among the three tested sham positions on the foot. Compared to the other tested sham-electrode setups, the highest MT RRs were achieved with the lowest pain perceptions and closest thresholds to the verum TTNS. In terms of electrode size, the 3.2 cm electrodes outperformed the 2.5 cm electrodes, giving a better RR; in fact, there was a 100% RR for both sensory and motor thresholds. The proposed sham configuration allows for the elicitation of sensation, and even some motor movement, without directly stimulating the tibial nerve.

### 3.3. First Patient Application

Over a phase of 6 weeks, a total of 15 verum TTNS applications and 14 sham stimulations were performed on the left leg of the patient, using the same standardised four-electrode setup. In addition to the two standard TTNS electrodes (Figure 2), two electrodes were placed on the 5th metatarsal (Figure 1), which was found to be the most suitable among the three sham positions under investigation. The patient tolerated the different types of interventions (comprising thresholding and intervention phase) well and did not report any pain or discomfort. The RRs for the sensory and motor responses were 100% for verum TTNS. Across all of the sham stimulations, the RR was 100% for sensory thresholds and 85% for MT. At the level of the motor response, the patient described the stimulation sensation commonly as tingling, prickling, pricking, or pressure for both verum and sham TTNS (Figure 6). The median CPT for verum TTNS was 11 mA (range: 8–13 mA); it was 22 mA (range: 6–34 mA) for the sham stimulation. The median motor threshold was 12.8 mA (range: 10.5–22.5 mA) for verum TTNS and 30 mA (range: 13.5–36 mA) for the sham stimulation.

## 4. Discussion

Stimulation at the 5th metatarsal position using 3.2 cm electrodes revealed the most suitable sham configuration for TTNS. Implemented as a four-electrode setup with standardized procedures, this appears to be a feasible RCT protocol for the envisaged patient population.

On average, all three sham positions showed rather higher threshold values compared to those of the verum TTNS, probably due to the fact that no nerves were directly stimulated in the sham positions. This indicates that stimulations at those positions should be safe and less likely to induce meaningful TTNS treatment effects. Regarding different sizes of electrodes, it is generally known that larger electrodes come with more comfortable stimulation; however, this is not a topic in the TNS literature and electrode size is rarely mentioned or discussed. Here, we looked into the differences between two slightly different electrode sizes (2.5 cm and 3.2 cm diameter, provided in the standard stimulator kit). Regarding all sensory and motor thresholds of the TTNS and sham stimulation applications, we did not find any significant effect resulting from electrode size. This was slightly unexpected but may be due to the rather small differences in area between the electrodes under investigation (8.0 cm^2^ vs. 4.9 cm^2^). It is therefore a very real possibility that we were simply not adequately powered to detect an effect resulting from electrode size. Similarly, the RR was always 100% for verum TTNS, irrespective of which electrodes were used. In contrast to this, higher RRs were achieved for sham MTs when using 3.2 cm compared to 2.5 cm electrodes. This means that irrespective of electrode size, it was always possible to position them in a way in which the tibial nerve could be successfully stimulated into inducing a detectable motor response. Regarding the sham positions, which, on purpose, were chosen to be somewhere remote from the target nerves, it makes sense that the smaller the electrodes were, the more difficult it became to induce any kind of motor response. A 100% motor RR could only be achieved using 3.2 cm electrodes at the 5th metatarsal sham position.

Regarding further comparisons among the different sham positions, significant effects were found regarding the plantar calcaneus, which had significantly higher sensory thresholds among all tested sham positions. The decreased sensory perception could be attributed to the anatomy of the plantar calcaneus sham site. The limited innervation and the presence of a thick layer of skin at this location may contribute to reduced sensation. In addition, impedance problems are more likely in this position, which can make stimulation difficult. In addition to electrode size and position, the perception of pain (NRS) was generally low in all tested positions. However, when using the 3.2 cm electrodes at the 5th metatarsal, there was a trend towards the lowest, and even lower, pain ratings than those of verum TTNS. In general, the sensory RR was 100% for all electrode sizes and positions, which could be attributed to our healthy study population without any known sensory deficits. This is likely to be different in the targeted patient population, with neurological deficits mostly associated with impaired sensation. The fact that the majority of patients with NLUTD suffer from impaired sensation is even advantageous when it comes to blinding maintenance. Potential differences between the sham and TTNS-induced sensations may be masked by the individually different sensory deficits.

Another argument for the choice of the 5th metatarsal position is that this position has already been used in the past in the context of sham-controlled TNS research [16,38]. Peters et al. even validated a corresponding sham protocol for percutaneous TNS in 2009 [38], where a sham stimulation was performed using two surface electrodes on the little toe. This was combined with a blunt needle to mimic the PTNS configuration and procedures, including causing sensation without stimulating the tibial nerve. To assess the maintenance of blinding, subjects filled in a questionnaire to assess the location and type of the sensation and to identify which leg received the sham stimulation and which one the PTNS. With only around one-third of correct guesses regarding sham stimulation, the conclusion from this study was that blinding was maintained. Apart from this study, there is a lack of firm statements on the maintenance of blinding in the TTNS literature, despite its relevance for sham-controlled TNS research.

In our recent pilot study, blinding was maintained by using a subsensory TTNS approach [23]; however, the piloting raised the question of whether the TTNS treatment dose was sufficient enough to yield clinical improvements. For this reason, this approach was not pursued further. A new approach with an increased TTNS dose that uses a stimulation intensity above the sensory threshold was considered. Obviously, this poses some challenges regarding the maintenance of blinding as the participants may perceive different stimulation procedures and sensations from different locations, comparing the verum to the sham TTNS. Considering our results presented above, stimulations at the 5th metatarsal position using a four-electrode setup additionally comprising two electrodes for verum TTNS may be an ideal sham configuration, mimicking the procedures and sensations of verum TTNS without meaningfully stimulating the tibial nerve. The advantage of this setup, obviously, is the opportunity for all patients to feel stimulation-induced sensations in the foot, which ideally are not easily distinguishable as verum or sham stimulations. This is particularly important for patients with preserved or close-to-normal sensations. However, the downside of this approach may be some risk toward persons in the sham group receiving a therapeutic treatment dose; this could reduce the treatment gap between the sham and verum groups. This could be reduced by stopping the sham stimulation after an initial short phase of threshold assessments; however, the participants may feel the cessation of the stimulation when it is stopped after some minutes. Default texts, translated to the respective languages of the study participants, need to be used to provide equal conditions for all participants regardless of their group allocation. By using the same device, as well as identical cables (matched outer appearance) for verum and sham stimulation, this can be ensured and will have beneficial effects for blinding maintenance. Blinding is crucial in order to control for nonspecific TTNS effects. These nonspecific effects could include placebo responses, natural variations in symptoms, and expectations of the patients. It is essential to carefully design studies and control for these factors to accurately assess the true therapeutic value of TNS. In addition, objective outcome measures (such as functional assessments, for instance, urodynamics, rather than self-reported questionnaires) are important to consider but cannot fully replace blinding measures.

Considering that participants may be familiar with the typical TTNS electrode placement, and even with the expected motor response of the big toe, further measures are required to optimise blinding. In our experience, once the view of the foot is restricted, it is rather difficult to tell whether and which toe is moving and how, particularly for individuals with impaired sensation. Therefore, each subjects’ view was obscured during the whole stimulation so that they were not able to see the toe movement. Furthermore, the subject positioning and the foot positioning should be standardized.

In a pilot study in patients with acute SCIs [12], the approach of stopping the sham stimulation after an initial threshold assessment was used. There, for both the verum and sham TTNS groups, the electrodes were placed at the same location: one electrode behind the medial malleolus and the other electrode 10cm superior. Although our verum electrode placement varies slightly, with the second electrode in the arch of the foot, such variations in verum electrode placement are irrelevant provided that sufficient tibial nerve stimulation is verified by a desired motor response (big toe flexion or fanning) [39]. In the verum TTNS group, stimulation was applied below the MT; in the sham group, the current was lowered to zero for the stimulation session after the motor response was verified. While this approach has the advantage of providing a comparable sensation during the initial determination of the thresholds, the cessation of the stimulation might be a potential risk factor in blinding maintenance. In addition, this approach comes with the risk that the patients in the sham group might receive sufficient enough stimulation to induce some TTNS effects (summed up over all of the sessions). Similarly, our sham approach of the active stimulation at the foot, albeit remote from the tibial nerve, bears the risk of some LUTS improvement in the sham group, potentially compromising the detectable differences when comparing to verum TTNS. This issue could be solved by refraining from applying any stimulation to the sham location [22]. However, such an approach comes at the cost of compromising blinding as the sham group was unable to perceive any stimulation or the presence of a motor response, which was in contrast to the verum TTNS experience. It can be concluded that our active sham approach at the 5th metatarsal inducing the same experience of sensation as in verum TTNS (with increasing current and the presence of a motor response) is suitable for sham-controlled blinded TTNS research in a clinical setting.

Depending on the target population, the sham group may be stimulated throughout the entire (30 min) treatment period. This bears the advantage of maintained blinding, particularly in patients with preserved sensation. However, the continued stimulation, albeit remote from the tibial nerve, might still have some therapeutic efficacy, especially after long-term stimulation. As an alternative, the stimulation intensity can be tapered or turned off for the actual treatment period. While this may enhance the treatment gap between verum and sham conditions, it could potentially compromise blinding in patients with intact sensory perception. This highlights the importance of tailoring the sham protocol to the specific application (e.g., self-administration or by healthcare professionals; at home or in the clinic; intermittent or sustained stimulation) and the patient population under investigation.

Another important criterion for maintaining the blinding and controlling for the nonspecific effects of TTNS is the time spent with the patient. Therefore, it is essential to establish a standardized procedure for both the sham and verum TTNS groups. As part of our methodological development, standardized instructions were provided to the participants in both groups. Furthermore, the preparation period encompassed consistent steps, including skin preparation, electrode placement, restriction of the subject’s view of the foot, and a thresholding phase, as outlined in corresponding standardized operating procedures. These collective factors contribute to a standardized process, particularly during the thresholding phase, which is designed to minimize patient’s ability to guess their group allocation. As a next step, it is crucial to investigate these aspects relevant to maintain blinding in both healthy volunteers and our target population.

One limitation of this study is that we did not assess any blinding information regarding verum and sham stimulation. Additionally, an active sham stimulation at the 5th metatarsal position may induce some clinical improvements, compromising the treatment gap between the verum and sham groups. Considering that, an ideal control protocol for TTNS would not stimulate the tibial nerve at all. However, in order to maintain blinding, the TTNS sham stimulation has to be carried out on the foot, in contrast to more remote stimulation sites and other nerves. It should also be noted that it is not known whether the stimulation of other nerves may also lead to some clinical improvements of LUTS. While neuromodulation involves afferent signal processing, the exact mechanism of action remains unclear. In patients with NLUTD, neuromodulation seems to target the dysfunctional nervous system by impacting the neuronal circuit patterns that are crucial to the proper functioning of the lower urinary tract. To gain more insights, additional research is warranted, including electrical sensory assessments and the examination of long-latency sensory evoked potentials specific to the lower urinary tract. Hence, there is a need for further investigations that explore the relevant factors and potential pathways associated with sensory functions in the lower urinary tract [40]. In addition, the current results were derived from healthy subjects that were younger compared to the envisaged patient cohorts. Motor RRs are generally believed to decrease with older age and neurological deficits. This is in line with our observation of the pilot patient, with a less than 100% motor RR for the sham TTNS. Furthermore, our sham protocol is limited to applications implemented by healthcare professionals and is not recommended for self-administration in double-blinded RCTs. Considering all these issues, we came up with a clearly defined protocol for the verum and sham TTNS to maintain blinding in neurological patients where sensory and motor responses might be impaired due to their neurological conditions.

The application of the newly developed sham-controlled four-electrode TTNS protocol was well-tolerated in one SCI patient over several weeks. The pilot patient reported no pain or discomfort during the verum and sham TTNS interventions. Regarding sensation, the patient described the sensations in the same manner when referring to verum and sham TTNS. When comparing the median CPTs and MTs of verum TTNS and sham interventions, similarly to the results in healthy subjects, higher CPTs and MTs were recorded for the sham group. This might be explained by the fact that we were not stimulating directly over a nerve in the sham compared to the verum TTNS. However, with only one patient measured, no conclusions can be drawn regarding thresholds at this stage. To sum up the first application in a patient: the four-electrode setup was feasible and well-tolerated over several weeks and it seems to be a promising approach that could be put to further use in upcoming RCTs to determine the specific effects of TTNS. Although TTNS has been investigated regarding patients with NLUTD by several authors [1,7,20], the results are difficult to compare due to variations in the study populations and stimulation protocols—particularly regarding the control configuration. Therefore, the evidence for TTNS effectiveness regarding NLUTD has yet to be demonstrated; new evidence should mostly be generated in blinded RCTs using our standardized four-electrode sham-controlled TTNS protocol.

## 5. Conclusions

Given a TTNS approach relative to motor instead of sensory thresholds, stimulation with 3.2cm electrodes at the 5th metatarsal position has proven to be the most suitable among the three tested sham-control configurations. It had the highest motor RR, lowest perceived pain level, and closest thresholds to verum TTNS in our healthy study population. Together with standardized blinding procedures, this can be used to assess neuromodulation-induced effects and distinguish them from nonspecific effects. For future RCTs in clinical settings, we suggest using the same four-electrode configuration (as described above) for sham and verum TTNS, according to a standardized protocol with predefined steps and instructions for the installation of the setup. Our results and considerations, together with the proposed optimized procedures, is a promising approach which should be further researched and implemented in future studies to increase our knowledge in the field of neuro-urology and neuromodulation.

## Figures and Tables

**Figure 1 biomedicines-11-01931-f001:**
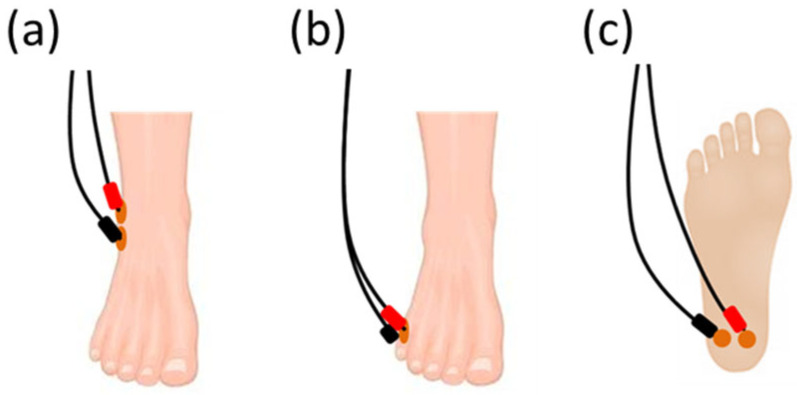
The three sham electrode positions under investigation using one pair of electrodes (**a**) at the lateral malleolus; (**b**) at the 5th metatarsal (below and on top of the little toe); (**c**) at the plantar calcaneus.

**Figure 2 biomedicines-11-01931-f002:**
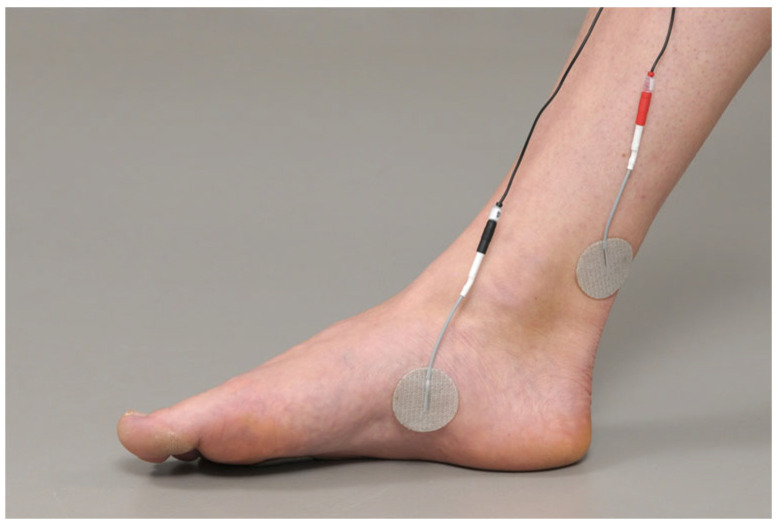
Standard TTNS setup using two surface electrodes (3.2 cm diameter).

**Figure 3 biomedicines-11-01931-f003:**
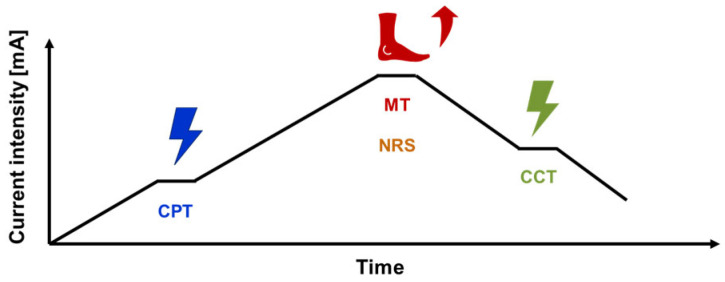
Thresholding procedure: gradual increase of stimulation intensity [mA] until the subject reports the current perception threshold (CPT); continued increase of stimulation intensity until first triggered movement (motor threshold (MT)) accompanied by a pain assessment (numerical rating scale (NRS) 0–10); gradual decrease of stimulation intensity until the subject reports the current cessation threshold (CCT).

**Figure 5 biomedicines-11-01931-f005:**
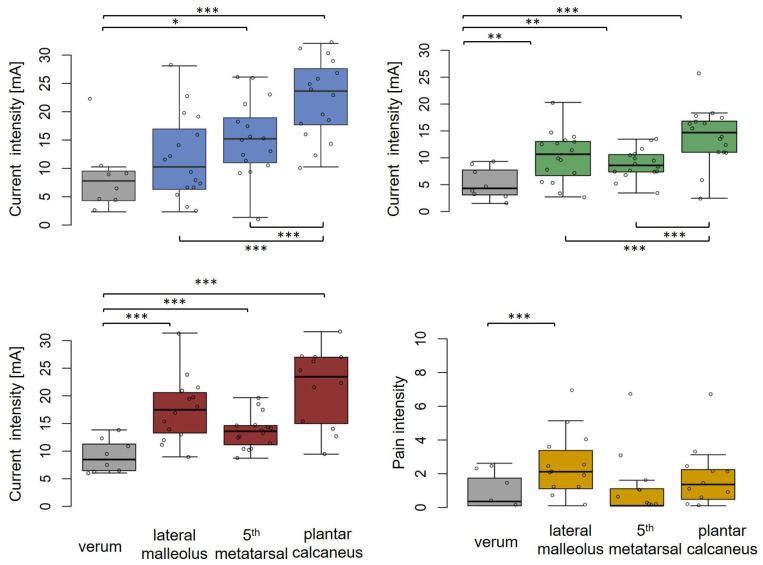
Boxplots for current perception threshold (CPT), current cessation threshold (CCT), and motor threshold (MT) with pain rating using a numeric scale (NRS). Verum (grey) and sham (coloured) values (n = 8) for 3.2 cm electrodes stratified for stimulation location. Significant differences were indicated by Wilcoxon signed-rank tests * (*p*-value < 0.05), ** (*p*-value < 0.01) *** (*p*-value < 0.001). Box–Whisker plot indications: lower whisker/upper whisker (maximum 1.5 × interquartile range), 25th percentile, 75th percentile, and median current intensities [mA] (CPT, CCT, and MT) or pain intensities (NRS).

**Figure 6 biomedicines-11-01931-f006:**
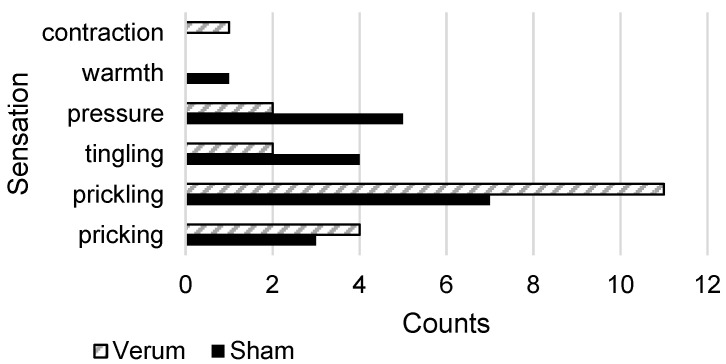
Subjective sensations reported by the patient for verum (hatched) and sham stimulations (multiple terms could be named per stimulation). The patient was asked to freely express her perception of the stimulation at the time of the occurrence of the motor response. Counts of the same responses are shown as bars per category for verum and sham stimulations.

**Table 1 biomedicines-11-01931-t001:** Response rate (%) for sensory (current perception thresholds (CPT), current cessation thresholds (CCT)) and motor thresholds (MT) in healthy subjects for the different electrode sizes and positions, including sham (n = 16) and verum TTNS (n = 8).

	Lateral Malleolus		5th Metatarsal		Plantar Calcaneus		Verum TTNS	
Electrode Size	2.5 cm	3.2 cm	2.5 cm	3.2 cm	2.5 cm	3.2 cm	2.5 cm	3.2 cm
CPT	100%	100%	100%	100%	100%	100%	100%	100%
MT	75%	86%	94%	100%	44%	75%	100%	100%
CCT	100%	100%	100%	100%	100%	100%	100%	100%

**Table 2 biomedicines-11-01931-t002:** Effect of electrode size (2) and sham position (3) on CPT, MT, CCT, and NRS score. Results from GLM regression models.

Setup Characteristics	CPT		MT		CCT		NRS	
	b [95% CI]	*p*-Value	b [95% CI]	*p*-Value	b [95% CI]	*p*-Value	b [95% CI]	*p*-Value
**Electrode Size**								
2.5 cm (reference)								
3.2 cm	0.88 [−0.18, 1.94]	0.104	0.55 [−2.16, 3.27]	0.690	1.14 [−0.68, 2.96]	0.220	−0.18 [−1.09, 0.73]	0.969
**Electrode Position**								
lateral malleolus (reference)								
5th metatarsal	0.89 [−0.24, 2.01]	0.123	−3.88 [−6.74, −1.02]	0.008	−1.05 [−3.06, 0.97]	0.307	−1.57 [−2.57, −0.58]	0.002
plantar calcaneus	3.32 [2.20, 4.45]	<0.001	2.21 [−1.03, 5.46]	0.181	3.74 [1.73, 5.76]	<0.001	−1.01 [−2.13, 0.12]	0.081

## Data Availability

The data presented in this article are available on request from the corresponding author.

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
