# Peer review of "Development of a Sham Protocol to Investigate Transcutaneous Tibial Nerve Stimulation in Randomised, Sham-Controlled, Double-Blind Clinical Trials"

_biomedicines, 2023, doi:10.3390/biomedicines11071931_

Round 1

Reviewer 1 Report

Thank you for allowing me to review this manuscript.  I believe it is of high quality and adds to the body of literature in the areas of neuromodulation and sham control trials.  Minor edits below:

Abstract:

19: “is limited due to inconsistent sham and blinding…”  This statement is false. I am very familiar with the TTNS body of literature, and I’ve asked myself, is it the inconsistent sham and blinding that limits the evidence? It is not.  It is the general lack of RCTs.

Line 26: (8/16 volunteers) is very confusing. What does this mean?

Body:

179: verum TTNS is different than what others have used, specifically regarding the placement of the second electrode which is placed proximal to the malleolus, like with PTNS and some of the references cited. It should be mentioned somewhere in this article that this is yet another area of variability in the TTNS literature- as well as current settings.

283: for people not familiar with the ISNSCI, the motor score should read 22/50 and the light touch and pin prick should read 98/112

336: this line should indicate that the sham is designed for the healthy control population, ie intact sensation. This would not be relevant to those with impaired sensation.

443:  The main issue with the sham stimulation selected is the motor response is quite different than verum. Why do the authors think that subjects will not be savvy enough to understand that toe flexion or fanning is the desired response?  The ideal sham would have electrodes in the same position as verum, and the current applied could achieve the verum motor response, but for sham would decrease to no current.  Perhaps in 2009, when PTNS was more novel, this would have been an adequate sham. But given the FDA approval and the many years of internet presence, it is not difficult to find the motor response to tibial nerve stimulation.  

473: the other limitations should be mentioned- lack of defined current settings, that is motor versus submotor. Also, lead configuration and likely frequency settings- I believe 10Hz and 20Hz have been used in the past.

488: I think another issue should be discussed.  The importance of blinding is greatly reduced when objective measures like urodynamics are used in studies.  Urodynamics don’t lie.

505-516: Again, it is important to mention that this is only a conclusion based on neurologically intact people. Considering the opportunity for TTNS to be studied in the neurologic population, it would be misleading to suggest this sham should be implemented in future studies in the neurologically impaired population.

Reviewer 2 Report

This manuscript compared the different design of sham protocol of transcutaneous tibial nerve stimulation (TTNS) in treatment of neurogenic lower urinary tract dysfunction (NLUTD). The authors found the suitable electrode size and position in the future sham treatment protocol. The study was well designed and scientifically merit. However, there are some critical points that authors should address.

1) Although sham treatment using a low amplitude of electrical stimulation, the patients still can perceive stimulation. This low stimulation might still have therapeutic efficacy after long-term stimulation. Therefore, the treatment should not be considered as sham treatment, but a low dose TTNS.

2) In patients with NLUTD, they might not perceive stimulation sensation from lower limb. It might not be necessary to use sham treatment by a low stimulation power.

3) The test sensory and motor threshold in healthy volunteers might not reflect the condition in patients with NLUTD.

4) TTNS has been widely applied in treatment of functional LUTD but not NLUTD. The authors might comment on the potential NLUTD for TTNS treatment.

Round 2

Reviewer 1 Report

From the author's response:

"Therefore, we restricted the view of participants, so that they were not able to see the toe movement. The approach with having the same two electrodes for sham and verum was piloted by our group (Tornic et al, 2020)."

If the subjects using TTNS can't see their foot, this is a terrible sham. Where in the manuscript does it say that subjects are blocked from viewing their toe movements? This is a very important aspect of the sham condition, especially considering that subjects would likely be performing TTNS at home on their own- they will have to see their toe movements.

There are clinical trials that happen in the lab/clinic that this sham would suffice, but this is not suitable for a home TTNS trial. This should be included in the conclusions.  People who know about tibial nerve stimulation appreciate that the advantage of TTNS is performing self neuromodulation at home. If restricting viewing of the foot is part of the sham, than this would not work for home trials, only clinic/lab trials.

Round 3

Reviewer 1 Report

Thank you for your attention to the comments and the edits. No further concerns.